# MOFDiff: Coarse-grained Diffusion for Metal–Organic Framework Design

**Xiang Fu**[1*†]    **Tian Xie**[2]    **Andrew S. Rosen**[3,4]    **Tommi Jaakkola**[1]    **Jake Smith**[2*]

[1]MIT CSAIL    [2]Microsoft Research AI4Science

[3]Department of Materials Science and Engineering, UC Berkeley

[4]Materials Science Division, Lawrence Berkeley National Laboratory

## Abstract

Metal–organic frameworks (MOFs) are of immense interest in applications such as gas storage and carbon capture due to their exceptional porosity and tunable chemistry. Their modular nature has enabled the use of template-based methods to generate hypothetical MOFs by combining molecular building blocks in accordance with known network topologies. However, the ability of these methods to identify top-performing MOFs is often hindered by the limited diversity of the resulting chemical space. In this work, we propose MOFDiff: a coarse-grained (CG) diffusion model that generates CG MOF structures through a denoising diffusion process over the coordinates and identities of the building blocks. The all-atom MOF structure is then determined through a novel assembly algorithm. As the diffusion model generates 3D MOF structures by predicting scores in E(3), we employ equivariant graph neural networks that respect the permutational and roto-translational symmetries. We comprehensively evaluate our model's capability to generate valid and novel MOF structures and its effectiveness in designing outstanding MOF materials for carbon capture applications with molecular simulations.

## 1   Introduction

Metal–organic frameworks (MOFs), characterized by their permanent porosity and highly tunable structures, are emerging as a versatile class of materials with applications spanning gas storage [24, 42], gas separations [44, 60], catalysis [77, 2, 64], and drug delivery [9, 39]. These frameworks are constructed from metal ions or clusters ("nodes") coordinated to organic ligands ("linkers"), forming a vast and diverse family of crystal structures [50]. Unlike traditional solid-state materials, MOFs offer unparalleled tunability, as their structure and function can be engineered by varying the choice of metal nodes and organic linkers. The surge in interest surrounding MOFs is evident in the increasing number of research studies dedicated to their synthesis, characterization, and computational design [36, 4, 80].

The modular nature of MOFs naturally lends itself to template-based representations and algorithmic assembly. These algorithms create hypothetical MOFs by connecting metal nodes and organic linkers (collectively, building blocks) along connectivity templates known as topologies [4, 76, 41]. Given a combination of topology, metal nodes, and organic linkers, the MOF structure is obtained through heuristic algorithms that arrange the building blocks, aligning them with the vertices and edges designated by the chosen topology, followed by a structural relaxation process based on classical force fields.

---

[*]Correspondence to Xiang Fu (`xiangfu@mit.edu`) and Jake Smith (`jakesmith@microsoft.com`).

[†]Work partially done during an internship at Microsoft Research AI4Science.

NeurIPS 2023 AI for Science Workshop.

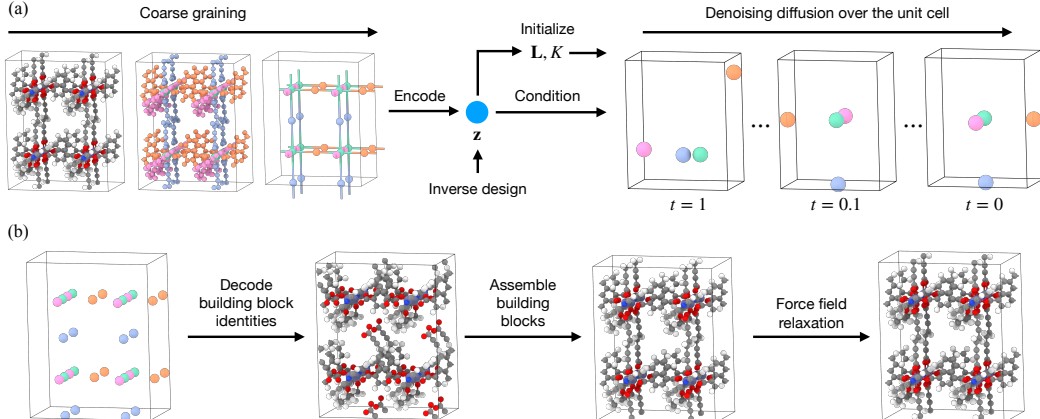

Figure 1: (a) MOFDiff encodes a coarse-grained (CG) representation of MOF structures and decodes CG MOF structures with a denoising diffusion process. To generate a coarse-grained MOF structure, the lattice parameters $L$ and the number of building blocks $K$ are predicted from the latent vector $z$ to initialize a random structure. A denoising diffusion process conditional on $z$ generates the building block identities and coordinates. Inverse design is enabled through gradient-based optimization over $z$ in the latent space. (b) The all-atom MOF structure is recovered from the coarse-grained representation through three steps: (1) the building block identities are decoded from the learned representation; (2) building block orientations are randomly initialized, then the assembly algorithm (Figure 7) is run to re-orient the building blocks; (3) the assembled structure goes through an energetic minimization process using the UFF force field. The relaxed structure is then used to compute structural and gas adsorption properties. Atom color code: Zn (purple), O (red), C (gray), N (blue), H (white).

The template-based approach to MOF design has led to the use of high-throughput computational screening approaches [6], variational autoencoders [78], genetic algorithms [16], Bayesian optimization [14], and reinforcement learning [81, 56] to discover new combinations of building blocks and topologies to identify top-performing materials. However, template-based methods enforce a set of pre-curated topology templates and building block identities. This inherently narrows the range of designs these hypothetical MOF construction methods can produce [51], possibly excluding materials suited for some applications. Therefore, we aim to derive a generative model based on 3D representations of MOFs without the need for pre-defined templates that often rely on chemical intuition.

Diffusion models [66, 28, 67] have made significant progress in generating molecular and inorganic crystal structures [65, 45, 73–75, 29, 33, 15, 30, 46, 79, 70, 40, 57, 32]. Recent work [55] also explored using a diffusion model to design linker molecules in specific MOFs. In terms of data characteristics, both inorganic crystals and MOFs are represented as atoms in a unit cell. However, a typical MOF unit cell contains hundreds of atoms (Figure 1), while the most challenging dataset studied in previous works [73, 47] only focused on inorganic crystals with less than 20 atoms in the unit cell. Training a diffusion model for complex MOF systems with atomic-scale resolution is not only technically challenging and computationally expensive but also suffers from extremely poor data efficiency. To name one challenge, without accounting for the internal structures of the metal clusters and the molecular linkers, directly applying diffusion models over the atomic representation of MOFs can very easily lead to unphysical structures for the inorganic nodes and/or organic linkers.

To address the challenges above, we propose MOFDiff, a coarse-grained diffusion model for generating 3D MOF structures that leverages the modular and hierarchical structure of MOFs (Figure 1 (a)). We derive a coarse-grained 3D representation of MOFs, a diffusion process over this CG MOF representation, and an assembly algorithm for recovering the all-atom MOF structure. In our experiments, we adapt the MOF dataset from Boyd et al. 6 (BW-DB) that contains hypothetical MOF structures and computed property labels related to separation of carbon dioxide ($CO_2$) from flue gas. We train MOFDiff on BW-DB and use MOFDiff to generate and optimize MOF structures for carbon capture.

In summary, the contributions of this work are:

- We derive a coarse-grained representation for MOFs where we specify the identities and coordinates of structural building blocks. We propose to learn a contrastive embedding to represent the vast building block design space.

- We formulate a diffusion process for generating coarse-grained MOF 3D structures. We then design an assembling algorithm that, given the identities and coordinates of building blocks, re-orients the building blocks to recover the atomic MOF structures. The generated atomic structures are further refined with force field relaxation (Figure 1 (b)).

- We demonstrate that MOFDiff can generate valid and novel MOF structures. MOFDiff surpasses the scope of previous template-based methods, producing MOFs that extend beyond simple combinations of pre-specified building blocks.

- We use MOFDiff to optimize MOF structures for carbon capture and evaluate the performance of the generated MOFs using molecular simulations. We show that MOFDiff can discover MOF structures with exceptional $CO_2$ adsorption properties with excellent efficiency.

## 2 Experiment Hightlight

Climate change is one of the most significant and urgent challenges that humanity needs to address. Carbon capture is one of the few technologies that can mitigate current $CO_2$ emissions, for which MOFs are promising candidate materials [69, 17]. In this experiment, we evaluate MOFDiff's capability to optimize MOF structures for use as $CO_2$-selective sorbents in point-capture applications. We train and evaluate our method on the BW-DB dataset. On average, each MOF contains 185 atoms (6.9 building blocks) in the unit cell. We highlight the results for optimizing carbon capture MOFs while deferring the results on unconditional sampling to Appendix D.

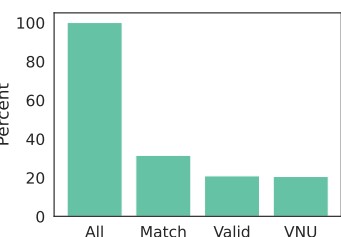

Figure 2: The validity of MOFD-iff samples optimized for $CO_2$ working capacity. "Match" stands for matched connection points. "VNU" stands for valid, novel, and unique. Definitions of the various validity criteria are included in Appendix D. Almost all valid samples are also novel and unique.

The key property for practical carbon capture purposes is high $CO_2$ working capacity, the net quantity of $CO_2$ capturable by a given quantity of MOF in an adsorption/desorption cycle. Several factors contribute to a high working capacity, such as the $CO_2$ selectivity over $N_2$, $CO_2/N_2$ uptake for each condition, and $CO_2$ heat of adsorption, which reflects the average binding energy of the adsorbing gas molecules. We compute these properties using molecular simulations, which is detailed in Appendix E.

**MOFDiff discovers promising candidates for carbon capture.**
We randomly sample 10,000 MOFs from the training dataset and encode these MOFs to get 10,000 latent vectors. We use the Adam optimizer [37] to maximize the model-predicted $CO_2$ working capacity for 5,000 steps with a learning rate of 0.0003. The resulting optimized latent vectors are then decoded, assembled, and relaxed. After conducting the validity checks described in Appendix D, we find 2054 MOFs that are valid, novel, and unique (Figure 3 (a)). These 2054 MOFs are then simulated with our GCMC workflow to compute gas adsorption properties. Given the systematic differences between the original labels of BW-DB and those calculated with our reimplemented GCMC workflow, we randomly sampled 5,000 MOFs from the BW-DB dataset and recalculated the gas adsorption properties using our GCMC workflow to provide a fair baseline for comparison. Figure 3 shows the $CO_2$ working capacity distribution of the BW-DB MOFs and the MOFDiff optimized MOFs: the MOFs generated by MOFDiff have significantly higher $CO_2$ working capacity. The four smaller panels break down the contributions to $CO_2$ working capacity from $CO_2/N_2$ selectivity, $CO_2$ heat of adsorption, as well as $CO_2$ uptake at the adsorption (0.15 bar, 298 K) and the desorption stages (0.1 bar, 363 K). We observe that MOFDiff generates a distribution of MOFs that are more selective towards $CO_2$, have higher $CO_2$ uptakes under adsorption conditions, and bind more strongly to $CO_2$.

From an efficiency perspective, GCMC simulations take orders of magnitude more computational time (tens of minutes to hours) than other components of the MOF design pipeline (seconds to tens of seconds). These simulations can also be made more accurate at significantly higher computational

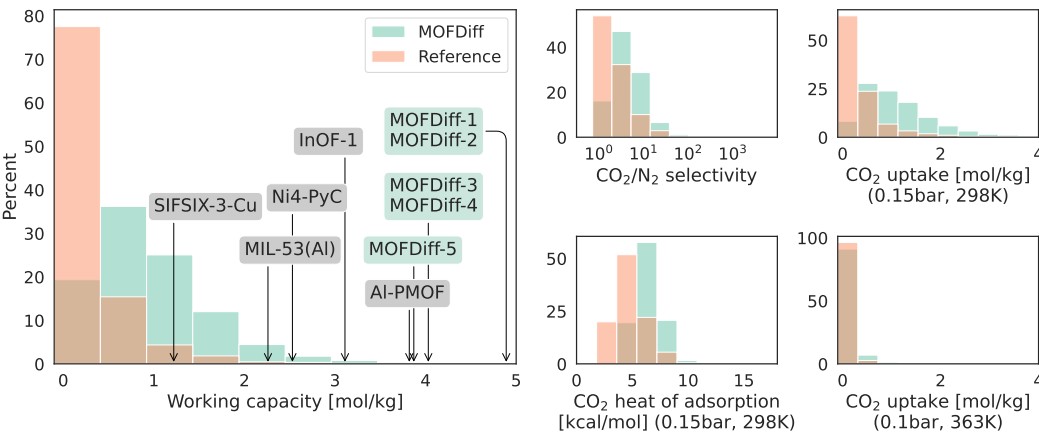

Figure 3: $CO_2$ adsorption properties for MOFDiff optimized samples (top-5 annotated with green boxes) compared to the reference distribution and selected MOF structures (grey boxes). The four small panels breakdown working capacity to more fundamental gas adsorption properties.

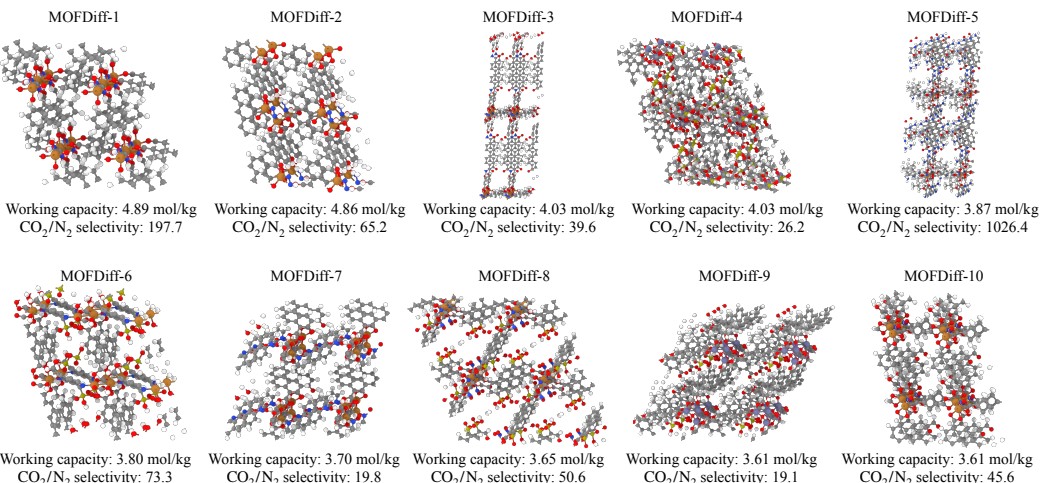

Figure 4: The top ten samples from MOFDiff in terms of the highest $CO_2$ working capacity. Atom color code: Cu (brown), Zn (purple), S (yellow), O (red), N (blue), C (gray), H (white).

costs (days) by converging sampling to tighter confidence intervals or using more advanced techniques, such as including blocking spheres, which prohibit Monte Carlo insertion of gas molecules into kinetically prohibited pores of the MOF, and calculating atomic charges with density functional theory (DFT). Therefore, the efficiency of a MOF design pipeline can be evaluated by the average number of GCMC simulations required to find one qualifying MOF for carbon capture applications. Naively sampling from the BW-DB dataset requires, on average, 58.1 GCMC simulations to find one MOF with a working capacity of more than 2 mol/kg. For MOFDiff, only 14.6 GCMC simulations are needed to find one MOF with a working capacity of more than 2 mol/kg, a 75% decrease in compute cost per candidate structure.

**Compare to carbon capture MOFs from literature.** Beyond efficiency, MOFDiff's generation flexibility also allows it to discover top MOF candidates that are outstanding for carbon capture. We compute gas adsorption properties of 18 MOFs that have been investigated for $CO_2$ adsorption from previous literature [48, 13, 25, 6] using our GCMC simulation workflow. We compare the gas adsorption properties of the top ten MOFs discovered from our 10,000 samples (visualized in Figure 4) to these 18 MOFs in Table 1 of Appendix E and annotate selected MOFs in Figure 3. MOFDiff can discover highly promising candidates, making up 9 out of the top 10 MOFs. In particular, Al-PMOF is

the top MOF selected by authors of Boyd et al. [6] from BW-DB. This comparison confirms MOFDiff's capability in advancing functional MOF design.

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

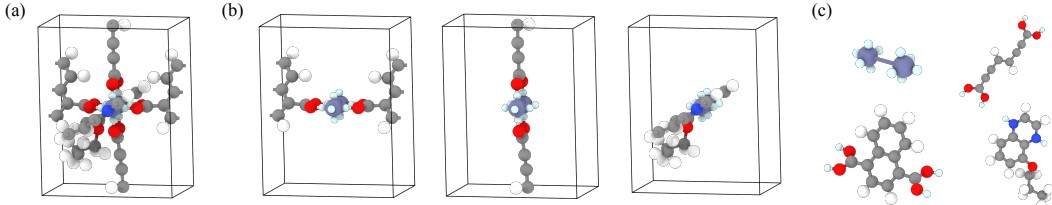

Figure 5: MOF decomposition with connections visualized. Connection points are light blue. (a) A MOF unit cell. (b) For visibility, we visualize the metal node and one other organic linker, one at a time. (c) All four building blocks in this example MOF.

# A    Representation of 3D MOF structures

Like any solid-state material, a MOF structure can be represented as the periodic arrangement of atoms in 3D space, defined by the infinite extension of a 3-dimensional unit cell. A unit cell that includes $N$ atoms is described by three components: (1) atom types $\boldsymbol{A} = (a_1, ..., a_N) \in \mathbb{A}^N$, where $\mathbb{A}$ denotes the set of all chemical elements; (2) atom coordinates $\boldsymbol{X} = (\boldsymbol{x}_1, ..., \boldsymbol{x}_N) \in \mathbb{R}^{N \times 3}$; and (3) periodic lattice $\boldsymbol{L} = (\boldsymbol{l}_1, \boldsymbol{l}_2, \boldsymbol{l}_3) \in \mathbb{R}^{3 \times 3}$. The periodic lattice defines the periodic translation symmetry of the material. Given $\boldsymbol{M} = (\boldsymbol{A}, \boldsymbol{X}, \boldsymbol{L})$, the infinite periodic structure is represented as,

$$\{(a_i', \boldsymbol{x}_i') | a_i' = a_i, \boldsymbol{x}_i' = \boldsymbol{x}_i + k_1 \boldsymbol{l}_1 + k_2 \boldsymbol{l}_2 + k_3 \boldsymbol{l}_3, k_1, k_2, k_3 \in \mathbb{Z}\}, \tag{1}$$

where $k_1, k_2, k_3$ are any integers that translate the unit cell using $\boldsymbol{L}$ to tile the entire 3D space. A MOF generative model aims to generate 3-tuples $\boldsymbol{M}$ that correspond to valid[3], novel, and functional MOFs. As noted in the introduction, prior research [73] employed a diffusion model on atomic types and coordinates to produce valid and novel inorganic crystal structures, specifically with fewer than 20 atoms in the unit cell. However, MOFs present a distinct challenge: their unit cells typically comprise tens to hundreds of atoms, composed of a diverse range of metal nodes and organic linkers. Directly applying the atomic diffusion model to MOFs poses formidable learning and computational challenges due to their increased size and complexity. This necessitates a new approach that can leverage the hierarchical nature of MOFs.

**Hierarchical representation of MOFs.** A coarse-grained 3D structural representation of a MOF can be derived from the coordinates and identities of the building blocks constituting the MOF. Such a representation is attractive, as the number of building blocks (denoted $K$) in a MOF is generally orders of magnitude smaller than the number of atoms (denoted $N$, $K \ll N$). We denote a coarse-grained MOF structure with $K$ building blocks as $\boldsymbol{M}^C = (\boldsymbol{A}^C, \boldsymbol{X}^C, \boldsymbol{L})$. The three components: (1) $\boldsymbol{A}^C = (a_1^C, ..., a_K^C) \in \mathbb{B}^K$ are the identities of the building blocks, where $\mathbb{B}$ denotes the set of all building blocks; (2) $\boldsymbol{X}^C = (\boldsymbol{x}_1^C, ..., \boldsymbol{x}_K^C) \in \mathbb{R}^{K \times 3}$ are the coordinates of the building blocks; (3) $\boldsymbol{L}$ are the lattice parameters. To obtain this coarse-grained representation, we need a systematic procedure to determine which atoms constitute which building blocks. In other words, we need an algorithm to assign the $N$ atoms to $K$ connected components, which correspond to $K$ building blocks.

Luckily, multiple methods have been developed for decomposing MOFs into building blocks based on network topology and MOF chemistry [7, 52, 3, 1, 43, 54]. We employ the `metal-oxo` algorithm from the popular MOF identification method `MOFid` [7]. Figure 1 (a) demonstrates the coarse-graining process: the atoms of each building block are identified with `MOFid` and assigned the same color in the visualization. From these segmented atom groups, we can compute the building block coordinates $\boldsymbol{X}^C$ and identities $\boldsymbol{A}^C$ for all $K$ building blocks to construct the coarse-grained representation. Each building block is extracted by removing single bonds that connect it to other building blocks. Every atom that forms such bonds to another building block is then assigned a special pseudo atom, called a connection point, at the midpoint of the original bonds that were removed. Figure 5 illustrates this process. We can now compute building block coordinates $\boldsymbol{X}^C$ by computing the centroid of the connection points for each building block[4].

---

[3]Assessing the validity of MOF 3D structures is hard in practice. We defer our protocol for validity determination to the experiment section.

[4]We compute the coarse-grained coordinates based on the connection points because the assembly algorithm introduced later relies on matching the connection points to align the building blocks.

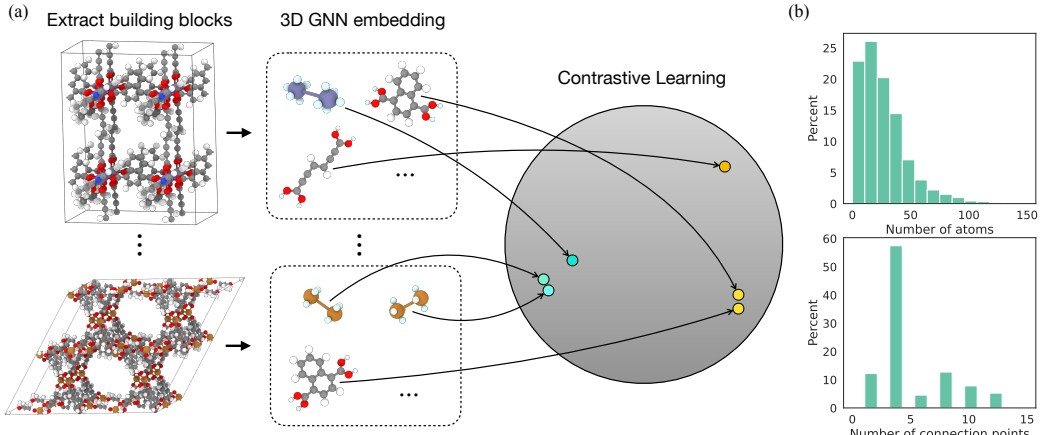

Figure 6: (a) Learning a compact representation of building blocks for CG diffusion. Building blocks are extracted from MOF structures and embedded through a GemNet-OC encoder. The representation is trained through a contrastive learning loss such that similar building blocks have similar embeddings. (b) The distribution of the number of atoms and the distribution of the number of connection points for the building blocks extracted from BW-DB. Atom color code: Cu (brown), Zn (purple), O (red), N (blue), C (gray), H (white).

The building block identities $\boldsymbol{A}^C$ are, on the other hand, tricky to represent because there is a huge space of possible building blocks for any non-trivial dataset. Furthermore, many building blocks share an identical chemical composition, varying only by small geometric variations in 3D orientation. Example building blocks are visualized in Figure 6 (a). To illustrate the vast space of building blocks, we extracted 2 million building blocks from the training split of the BW-DB dataset (289k MOFs). To quantify the extent of geometric variation among building blocks with the same molecule/metal cluster, we computed the ECFP4 fingerprints [63] for each building block using their molecular graphs and found 242k unique building block identities. This building block space is too large to be represented as a categorical variable in a generative model.

**Contrastive representation of building blocks.** In order to construct a compact representation of building blocks for diffusion-based modeling, we use a contrastive learning approach [26, 11] to embed building blocks into a low dimensional latent space. A building block $i$ is encoded as a vector $\boldsymbol{b}_i$ using a GemNet-OC encoder [22, 23], an E(3)-invariant graph neural network model. We then train the GNN building block encoder using a contrastive loss to map small geometric variations of the same building block to similar latent vectors in the embedding space. In other words, two building blocks are a positive pair for contrastive learning if they have the same ECFP4 fingerprint. Figure 6 (a) illustrates the contrastive learning process, while Figure 6 (b) shows the distribution of the number of atoms and the distribution of the number of connection points for building blocks extracted from BW-DB. The contrastive loss is defined as:

$$\mathcal{L}_{\mathrm{C}} = -\log \sum_{i \in \boldsymbol{B}} \frac{\sum_{j \in \boldsymbol{B}_i^+} \exp(s_{i,j}/\tau)}{\sum_{j \in \boldsymbol{B}} \exp(s_{i,j}/\tau)} \tag{2}$$

where $\boldsymbol{B}$ is a training batch, $\boldsymbol{B}_i^+$ are the other data points in $\boldsymbol{B}$ that have the same ECFP4 fingerprint as $i$, $s_{i,j}$ is the similarity between building block $i$ and building block $j$, and $\tau$ is the temperature factor. We define $s_{i,j} = \boldsymbol{p}_i^T \boldsymbol{p}_j / (||\boldsymbol{p}_i|| ||\boldsymbol{p}_j||)$, which is the cosine similarity between projected embeddings $\boldsymbol{p}_i$ and $\boldsymbol{p}_j$. The projected embedding is obtained by projecting the building block embedding $\boldsymbol{b}_i$ using a multi-layer perceptron (MLP) projection head: $\boldsymbol{p}_i = \mathrm{MLP}(\boldsymbol{b}_i)$. The projection layer is a standard practice in contrastive learning frameworks for improved performance.

With a trained building block encoder, we encode all building blocks extracted from a MOF to construct the building block identities in the coarse-grained representation: $\boldsymbol{A}^C = (\boldsymbol{b}_1, ..., \boldsymbol{b}_K) \in \mathbb{R}^{K \times d}$, where $d$ is the embedding dimension of the contrastive building block encoder ($d = 32$ for BW-DB). The contrastive embedding allows accurate retrieval through finding the nearest neighbor in the embedding space.

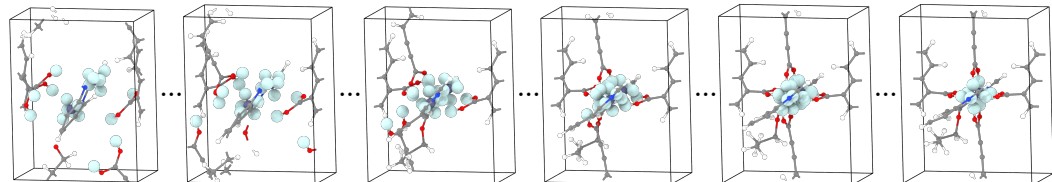

Figure 7: The MOF assembly process. Connection points (light blue) are highlighted for visibility.

**Building block representation.** The building block encoder is a GemNet-OC model that inputs the 3D configuration of the building block, including the connection points, and outputs building block embedding $\boldsymbol{b}$. A radius-cutoff graph is built as the building block for message passing. In addition to the contrastive loss $\mathcal{L}_C$, we also train the building block latent representation to encode the number of atoms $N_b$, the number of connection points $C_b$, and the largest distance between any pair of atoms $l$ in the building block by predicting these quantities. Cross-entropy loss is used for $N_b$ and $C_b$, while mean squared error loss is used for $l$:

$$\mathcal{L}_{\mathrm{B}} = \mathrm{CrossEntropy}(N_b, \hat{N}_b) + \mathrm{CrossEntropy}(C_b, \hat{C}_b) + ||l - \hat{l}||^2 \qquad (3)$$

where $\hat{N}_b, \hat{C}_b, \hat{l}$ are model predictions. These quantities are important indicators of the size and connection pattern of the building block. The overall loss for the building block encoder is:

$$\mathcal{L}_{\mathrm{BB}} = \mathcal{L}_{\mathrm{C}} + \mathcal{L}_{\mathrm{B}} + \beta_{\boldsymbol{b}}||\boldsymbol{b}||^2 \qquad (4)$$

where the last term is an $L_2$ regularization over the building block embedding with a loss weighting of $\beta_{\boldsymbol{b}} = 0.0001$ to constrain the norm of building block embedding. The regularization makes the embedding numerically stable to use in diffusion modeling later. We do not apply weighting over $\mathcal{L}_{\mathrm{C}}$ and $\mathcal{L}_{\mathrm{B}}$. Hyperparameters of the building block encoder are reported in Table 2. GemNet-OC hyperparameters are the default values for the Base version from Gasteiger et al. [23] unless otherwise noted. After being trained to convergence, the building block encoder is frozen and used for encoding all building blocks to construct the CG representation of MOFs.

## B    MOF design with coarse-grained diffusion

**MOFDiff.** Equipped with the CG MOF representation, we encode MOFs as latent vectors and decode MOF structures with conditional diffusion. The MOFDiff model is composed of four components (Figure 1 (a)): (1) A periodic GemNet-OC encoder[5] that outputs a latent vector $\boldsymbol{z} = \mathrm{PGNN}_{\mathrm{E}}(\boldsymbol{M}^C)$; (2) an MLP predictor that predicts the lattice parameters and the number of building blocks from the latent code $\boldsymbol{z}$: $\hat{\boldsymbol{L}}, \hat{K} = \mathrm{MLP}_{\boldsymbol{L}, K}(\boldsymbol{z})$; (3) a periodic GemNet-OC denoiser that denoises random structures to CG MOF structures conditional on the latent code: $\boldsymbol{s}_{\boldsymbol{A}^C}, \boldsymbol{s}_{\boldsymbol{X}^C} = \mathrm{PGNN}_{\mathrm{D}}(\tilde{\boldsymbol{M}}_t^C, \boldsymbol{z})$, where $\boldsymbol{s}_{\boldsymbol{A}^C}, \boldsymbol{s}_{\boldsymbol{X}^C}$ are the predicted scores for building block identities $\boldsymbol{A}^C$ and coordinates $\boldsymbol{X}^C$, and $\tilde{\boldsymbol{M}}_t^C$ is a noisy CG structure at time $t$ in the diffusion process; (4) an MLP predictor that predicts properties $\boldsymbol{c}$ (such as $CO_2$ working capacity) from $\boldsymbol{z}$: $\hat{\boldsymbol{c}} = \mathrm{MLP}_{\mathrm{P}}(\boldsymbol{z})$.

The first three components are used to generate MOF structures, while the property predictor $\mathrm{MLP}_{\mathrm{P}}$ can be used for property-driven inverse design. To sample a CG MOF structure from MOFDiff, we follow three steps: (1) randomly sample a latent code $\boldsymbol{z} \sim \mathcal{N}(\boldsymbol{0}, \boldsymbol{I})$; (2) decode the lattice parameters $\boldsymbol{L}$ and the number of building blocks $K$ from $\boldsymbol{z}$, use $\boldsymbol{L}$ and $\boldsymbol{z}$ to initialize a random coarse-grained MOF structure $\tilde{\boldsymbol{M}}^C = (\tilde{\boldsymbol{A}}^C, \tilde{\boldsymbol{X}}^C, \boldsymbol{L})$; (3) generate the coarse-grained MOF structure $\boldsymbol{M}^C = (\boldsymbol{A}^C, \boldsymbol{X}^C, \boldsymbol{L})$ through the denoising diffusion process conditional on $\boldsymbol{z}$. Given the final building block embedding $\boldsymbol{A}^C \in \mathbb{R}^{K \times d}$, we decode the building block identities by finding the nearest neighbors in the building block embedding space of the training set.

**Recover all-atom MOF structures.** The orientations of building blocks are not specified by the CG MOF representation, but they can be determined by forming connections between the building blocks. We design an assembly algorithm that optimizes the building block orientations to match the

---

[5]We refer interested readers to Xie & Grossman [72], Chen et al. [10], Xie et al. [73] for details about handling periodicity in graph neural networks.

connection points of adjacent building blocks such that the MOF becomes connected (visualized in Figure 7). This optimization algorithm places Gaussian densities at the position of each connection point and maximizes the overlap of these densities between compatible connection points. Two connection points are compatible if they come from two different building blocks: one is from a metal atom, and the other is from a non-metal atom (Figure 5). The radius of the Gaussian densities is gradually reduced in the optimization process: at the beginning, the radius is high, so the optimization problem is smoother, and it is simpler to find an approximate solution. At the end of optimization, the radius of the densities is small, so the algorithm can find accurate orientations for matching the connection points closely. This overlap-based loss function is differentiable with regard to the building block orientation, and we optimize for the building block orientations using the L-BFGS optimizer [8].

The assembly algorithm outputs an all-atom MOF structure that is fed to a structural relaxation procedure using the UFF force field [62]. We modify a relaxation workflow from previous work [53] implemented with LAMMPS [68] and LAMMPS Interface [5] to refine both atomic positions and the lattice parameters using the conjugate gradient algorithm.

**Full generation process.** Six steps are needed to generate a MOF structure: (1) sample a latent vector $z$; (2) decode the lattice parameters $L$ and the number of building blocks $K$ from $z$, use $L$ and $K$ to initialize a random coarse-grained MOF structure; (3) generate the coarse-grained MOF structure through the denoising diffusion process conditional on $z$; (4) decode the building block identities by finding their nearest neighbors from the building block vocabulary; (5) use the assembly algorithm to re-orient building blocks such that compatible connection points' overlap is maximized; (6) relax the all-atom structure using the UFF force field to refine the lattice parameter and atomic coordinates. All steps are demonstrated in Figure 1.

## C  Model Details

**MOFDiff encoding.** Before feeding the MOF structures to the periodic GNN encoder, we normalize all MOFs by dividing all lattice lengths by the mean lattice length and dividing all building block embedding by the mean of all building block embedding's $L_2$–norms. This normalization makes it easier to select the noisy distributions for diffusion modeling. The coarse-grained diffusion model only operates on the coarse-grained representation. To encode a CG MOF structure, we build the coarse-grained graph with the CG connections inferred from the all-atom inter-building-block connections: two building blocks $i$ and $j$ have an edge with the periodic image $I$ if an atom in building block $i$ has a bond connection to an atom in building block $j$ (considering periodic image $I$). We refer interested readers to Xie et al. 73 for more details on the multi-graph representation of crystals. The periodic GNN encoder is an E(3)-invariant GemNet-OC model. After invariant message passing, we apply pooling to the node embedding to obtain the CG MOF latent code $z$.

**Diffusion process.** The forward diffusion process injects noise into the coarse-grained MOF $M^C = A^C, X^C, L$) to obtain the noisy structure $\tilde{M}_t^C = (\tilde{A}_t^C, \tilde{X}_t^C, L)$ for $t = 0$ to $T$, where at $t = T$ the data is diffused to the prior distribution. At time step $t$, the denoiser $\text{PGNN}_D$ inputs the noisy structure $M_t^C$, latent code $z$, and the time step $t$ then predicts scores $s_{A_t^C, z}, s_{X_t^C, z}$ for building block embedding and coordinates. The lattice parameter remains fixed throughout the diffusion process. With contrastive building block embedding in $\mathbb{R}^{K \times d}$ ($d = 32$ for BW-DB), we employ a DDPM [28] (variance-preserving) forward process for type embedding:

$$q(\tilde{A}_t^C | \tilde{A}_{t-1}^C) = \mathcal{N}(\sqrt{1 - \beta_t} \cdot \tilde{A}_{t-1}^C, \beta_t I) \tag{5}$$

$$q(\tilde{A}_t^C | \tilde{A}_0^C) = \mathcal{N}(\sqrt{\bar{\alpha}_t} \cdot \tilde{A}_0^C, (1 - \bar{\alpha}_t)I) \tag{6}$$

where $\beta_1, \ldots, \beta_T$ is the variance schedule, $\alpha_t := 1 - \beta_t$ and $\bar{\alpha}_t = \prod_{s=1}^t \alpha_s$. The corresponding reverse diffusion sampling process is:

$$q(\tilde{A}_{t-1}^C | \tilde{M}_t^C, z) = \mathcal{N}\left(\frac{1}{\sqrt{\alpha_t}}\left(A_t^C - \frac{1 - \alpha_t}{\sqrt{1 - \bar{\alpha}_t}} s_{A_t^C, z}\right), \frac{1 - \bar{\alpha}_{t-1}}{1 - \bar{\alpha}_t}\beta_t I\right) \tag{7}$$

We refer interested readers to Ho et al. [28] for a more detailed derivation of the DDPM diffusion process. We use the same noise schedule as Hoogeboom et al. [29], a, for the building block type diffusion.

With building block coordinates in $\mathbb{R}^{K \times 3}$, we employ a variance-exploding forward diffusion process for the coordinates:

$$q(\tilde{\boldsymbol{X}}_t^C | \tilde{\boldsymbol{X}}_0^C) = \mathcal{N}(\tilde{\boldsymbol{X}}_0^C, \sigma_t^2 \boldsymbol{I}) \tag{8}$$

where $\sigma_1, \ldots, \sigma_T$ are noise levels. The corresponding reverse diffusion sampling process is:

$$q(\tilde{\boldsymbol{X}}_{t-1}^C | \tilde{\boldsymbol{M}}_t^C, \boldsymbol{z}) = \mathcal{N}\left(\tilde{\boldsymbol{X}}_t^C - \sqrt{\sigma_t^2 - \sigma_{t-1}^2} \cdot \boldsymbol{s}_{\boldsymbol{X}_t^C, \boldsymbol{z}}, \frac{\sigma_{t-1}^2(\sigma_t^2 - \sigma_{t-1}^2)}{\sigma_t^2} \boldsymbol{I}\right) \tag{9}$$

We refer interested readers to Song et al. [67] for a more detailed derivation of the variance-exploding diffusion process. We use the same noise schedule as Song et al. [67]: $\sigma_t = \sigma_{\min}\left(\frac{\sigma_{\max}}{\sigma_{\min}}\right)^{\frac{t-1}{T-1}}$. We handle the denoising target under periodicity similarly as Xie et al. [73] and direct readers interested in further details to this reference.

To train the denoising score network $\mathrm{PGNN}_D$, we use the following loss functions:

$$\mathcal{L}_{\boldsymbol{A}} = \mathbb{E}_{t,\boldsymbol{M}^C,\boldsymbol{\epsilon_A}}\left[||\boldsymbol{\epsilon_A} - \boldsymbol{s}_{\boldsymbol{A}_t^C, \boldsymbol{z}}||^2\right] \quad \text{and} \quad \mathcal{L}_{\boldsymbol{X}} = \mathbb{E}_{t,\boldsymbol{M}^C,\boldsymbol{\epsilon_X}}\left[\sigma_t^2 ||\boldsymbol{\epsilon_X} - \boldsymbol{s}_{\boldsymbol{X}_t^C, \boldsymbol{z}}||^2\right] \tag{10}$$

where $\boldsymbol{\epsilon_A}, \boldsymbol{\epsilon_X} \sim \mathcal{N}(\boldsymbol{0}, \boldsymbol{I})$ are sampled Gaussian noises, injected through the forward diffusion processes defined in Equation (6) and Equation (8). The reverse diffusion process defined in Equation (7) and Equation (9) are used for sampling MOF structures at inference time.

In addition to the diffusion losses $\mathcal{L}_{\boldsymbol{A}}$ and $\mathcal{L}_{\boldsymbol{X}}$, MOFDiff is also trained to predict the lattice parameters $\hat{\boldsymbol{L}}$, the number of building blocks $\hat{K}$ and property labels $\hat{\boldsymbol{c}}$ from the latent code $\boldsymbol{z}$. We use a mean squared error loss for the lattice parameters and the property labels, and a cross-entropy loss for the number of building blocks:

$$\mathcal{L}_{\boldsymbol{L},K,\boldsymbol{c}} = ||\boldsymbol{L} - \hat{\boldsymbol{L}}||^2 + \mathrm{CrossEntropy}(K, \hat{K}) + ||\boldsymbol{c} - \hat{\boldsymbol{c}}||^2 \tag{11}$$

The entire MOFDiff is then trained end-to-end with the loss function:

$$\mathcal{L}_{\mathrm{MOFDiff}} = \mathcal{L}_{\boldsymbol{A}} + \mathcal{L}_{\boldsymbol{X}} + \mathcal{L}_{\boldsymbol{L},K,\boldsymbol{c}} + \beta_{\mathrm{KL}}\mathcal{L}_{\mathrm{KL}} \tag{12}$$

Where the $\mathcal{L}_{\mathrm{KL}}$ is the KL regularization for variational autoencoders. We did not use weighting over the different loss terms except for the KL regularization, which is weighted with $\beta_{\mathrm{KL}} = 0.01$ [27]. All hyperparameters are reported in Table 3.

The coarse-grained MOF structures generated by the diffusion model specify the lattice parameters, building block identities, and building block coordinates (the centroid of connection points). However, they do not specify the orientations of building blocks. The assembly algorithm finds the orientations of the building blocks to connect them to each other. Throughout the assembly process, we fix the centroids of the building blocks, the internal structures (atom relative coordinates) of the building blocks, and the lattice parameters. The building block orientations are the only variables that are allowed to change (Figure 7). As we change the orientation of a building block, all atoms and connection points within rotate around its centroid.

For any ground truth structure, the connection points (as defined in Appendix A) of adjacent building blocks will perfectly overlap since they are midpoints of the bonds connecting inter-building-block atoms. Therefore, a viable objective for the assembly algorithm is to maximize the overlap of compatible inter-building-block connection points. Two connection points are compatible if (1) one connection point is from a metal atom, and the other is from a non-metal atom; (2) they are not from the same building block. We denote the set of all connection points as $\boldsymbol{C}$, the number of connection points as $C$, the coordinate of connection point $i$ as $\boldsymbol{x}_i$, the Euclidean distance between connection points $i$ and $j$ as $d_{ij} := ||\boldsymbol{x}_i - \boldsymbol{x}_j||$, and the connection points compatible with $i$ as $\boldsymbol{C}_i$. We define the objective function:

$$\mathcal{L}_{\mathrm{O},k,\sigma} = -\frac{1}{C} \sum_{i \in \boldsymbol{C}} \sum_{j \in \boldsymbol{C}_i} \exp(\frac{-d_{ij}}{\sigma^2}) \cdot \mathbb{I}(||\{q : q \in \boldsymbol{C}_i, d_{iq} \leq d_{ij}\}|| \leq k) \tag{13}$$

---

**Algorithm 1** Optimize building block orientations for MOF assembly

---

1: **Input**: MOF structure $\boldsymbol{M} = (\boldsymbol{A}^C, \boldsymbol{X}^C, \boldsymbol{L})$, the number of optimization rounds $U$, Gaussian kernel width $\sigma_1 > \cdots > \sigma_U$, number of nearest neighbors for overlap evaluation $k_1 > \cdots > k_U$

2: **Output**: Building block orientations: $\boldsymbol{\Omega} = \left\{ \boldsymbol{\omega}_{a_i^C} \text{ for all } a_i^C \in \boldsymbol{A}^C \right\}$

3: Randomly initialize building block orientations $\boldsymbol{\Omega}$

4: **for** round $u = 1, \ldots, U$ **do**

5:     Let $\sigma \leftarrow \sigma_u$, $k \leftarrow k_u$

6:     minimize $\mathcal{L}_{\text{O},k,\sigma}(\boldsymbol{\Omega})$ with respect to $\boldsymbol{\Omega}$ using L-BFGS

7: **end for**

---

Where $\mathbb{I}$ is the indicator function. This loss can be thought of as measuring the inverse of the overlap under a Gaussian kernel of width $\sigma$, and the overlap is only evaluated for the $k$ nearest neighbors among the compatible connection points. Minimizing this loss maximizes the overlap. This loss is related to the building block orientations because the coordinate of a connection point $\boldsymbol{x}_i$ is related to the orientation $\boldsymbol{\omega}_a$ (under the axis-angle representation) and CG coordinate $\boldsymbol{x}_a^C$ of the corresponding building block $a$ through:

$$\boldsymbol{x}_i = \boldsymbol{x}_a^C + \boldsymbol{v}_{a,i} \boldsymbol{R}_a \tag{14}$$

where $\boldsymbol{v}_{a,i}$ is the vector from the building block centroid to the connection point under a canonical orientation (which is invariant throughout the assembly process), and $\boldsymbol{R}_a$ is the rotation matrix corresponding to $\boldsymbol{\omega}_a$. The distance between a pair of connection points $d_{ij}$ can then be related to the orientations of the two corresponding building blocks through Equation (14). $\mathcal{L}_{\text{O},k,\sigma}$ is twice-differentiable with respect to building block rotations $\boldsymbol{\omega}$ for all building blocks as $\mathcal{L}_{\text{O},k,\sigma}$ is twice-differentiable with respect to $d_{ij}$ for all connection points $i, j$, and $d_{ij}$ is twice-differentiable with respect to $\boldsymbol{\omega}_a$ and $\boldsymbol{\omega}_b$. This allows us to use L-BFGS, a second-order optimization algorithm.

We can now define an annealed optimization process by gradually reducing $\sigma$ and $k$: at the beginning, the width $\sigma$ and the number of other connection points we evaluate overlap with $k$ are high, so it is easier to find overlap between connection points, and the optimization problem becomes smoother. This makes it simpler to find an approximate solution. At the end of optimization, the kernel width $\sigma$ is small, and we are only computing the overlap for the closest compatible connection points. At this stage, the algorithm should have already found an approximate solution, and a stricter evaluation over overlapping can let the algorithm find more accurate orientations for matching the connection points closely.

The assembly algorithm starts by randomly initializing the orientations of the building blocks. Using the L-BFGS method, the algorithm iteratively minimizes $\mathcal{L}_{\text{O},k,\sigma}$ by adjusting the building block orientations $\boldsymbol{\Omega}$: $\boldsymbol{\omega}_{a_i^C}$ (using the axis-angle representation) for all building blocks $a_i^C$. We use the axis-angle representations because rotation matrices need to follow specific constraints. As explained above, we start with a relatively high $\sigma$ and $k$ and gradually reduce them in the optimization process to gradually refine the optimized orientations. The full algorithm is shown in Algorithm 1. In our experiments, we use 3 rounds: $U = 3$, with $\sigma = [3, 1.65, 0.3]$ and $k = [30, 16, 1]$. An example assembly process is visualized in Figure 7.

**Force field relaxation.** The relaxation process is modified from a workflow proposed in Nandy et al. 52 and has four rounds of energy minimization using the UFF force field and the conjugate gradient algorithm in LAMMPS. At each round, we use LAMMPS's minimize function with `etol`$=1 \times 10^{-8}$, `ftol`$=1 \times 10^{-8}$, `maxiter`$=1 \times 10^6$, and `maxeval`$=1 \times 10^6$. In the first and third rounds, we only relax the atom coordinates while keeping the lattice parameters frozen. In the second and fourth rounds, we relax both atom coordinates and the lattice parameters. The relaxation process can refine the all-atom structures based on the complete MOF configuration and correct minor errors in the previous steps (such as slightly smaller/bigger unit cells). Structural optimization using classical force field is commonly done in materials and MOF design [41, 52].

## D    Generate valid and novel MOF structures

**Determine the validity and novelty of MOF structures.** Assessing MOF validity is generally challenging. We employ a series of validity checks:

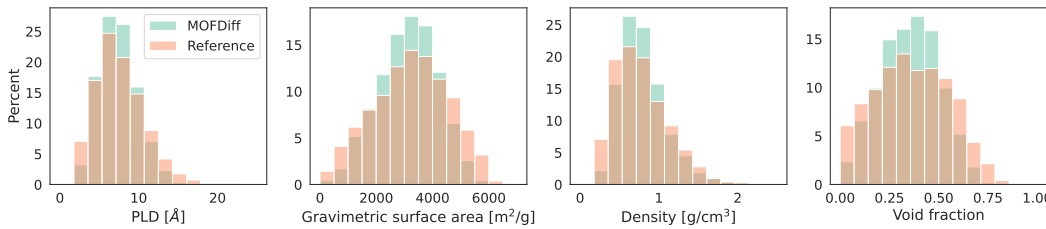

Figure 8: MOFDiff samples match the reference distribution for various structural properties.

1. The number of metal connection points and the number of non-metal connection points should be equal. We call this criterion Matched Connection.

2. The MOF atomic structure should successfully converge in the force field relaxation process.

3. For the relaxed structure, we adopt `MOFChecker` [31] to check validity. `MOFChecker` includes a variety of criteria: the presence of metal and organic elements, porosity, no overlapping atoms, no non-physical atomic valences or coordination environments, no atoms or molecules disconnected from the primary MOF structure, and no excessively large atomic charges. We refer interested readers to Jablonka 31 for details.

We say a MOF structure is **valid** if all three criteria above are satisfied. For novelty, we adopt the MOF identifier extracted by `MOFid` and say a MOF is **novel** if its `MOFid` differs from any other MOFs in the training dataset. We also count the number of **unique** generations by filtering out replicate samples using their `MOFid`. We are ultimately interested in the valid, novel, and unique (VNU) MOFs discovered.

**MOFDiff generates valid and novel MOFs.** A prerequisite for functional MOF design is the capability to generate novel and valid MOF structures. We randomly sample 10,000 latent vectors from $\mathcal{N}(\mathbf{0}, \mathbf{I})$, decode through MOFDiff, assemble, and apply force field relaxation to obtain the atomic structures. Figure 9 shows the number of MOFs satisfying the validity and novelty criteria: out of the 10,000 generations, 5,865 samples satisfy the matching connection criterion; 3012 samples satisfy the validity criteria, and 2998 MOFs are valid, novel, and unique. To evaluate the structural diversity of the MOFDiff samples, we investigate the distribution of four important structural properties calculated with `Zeo++` [71]: the diameter of the smallest passage in the pore structure, or pore limiting diameter (PLD); the surface area per unit mass, or gravimetric surface area; the mass per unit volume, or density; and the ratio of total pore volume to total cell volume, or void fraction [49]. These structural properties, which characterize the geometry of the pore network within the MOF, have been shown to correlate directly with important properties of the bulk material [38]. The distributions of MOFDiff samples and the reference distribution of BW-DB are shown in Figure 8. We observe that the property

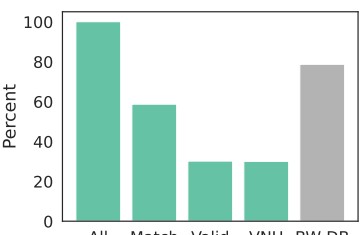

Figure 9: The validity of MOFDiff samples for increasingly strict criteria. "Match" stands for matched connection. "VNU" stands for valid, novel, and unique. Almost all valid samples are also novel and unique. The last column shows the validity percentage of BW-DB under our criteria.

distribution of generated samples matches well with the reference distribution of BW-DB, covering a wide range of property values.

## E   Experiment Details

The BW-DB dataset contains 304k MOFs with less than 20 building blocks (as defined by the `metal-oxo` decomposition algorithm) from the 324k MOFs in Boyd et al. 6. We limit the size of MOFs within the dataset under the hypothesis that MOFs with extremely large primitive cells may be difficult to synthesize. The median lattice constant in the primitive cell of an experimentally realized MOF in the Computation-Ready, Experiment (CoRE) MOF 2019 dataset is, for example, only 13.8

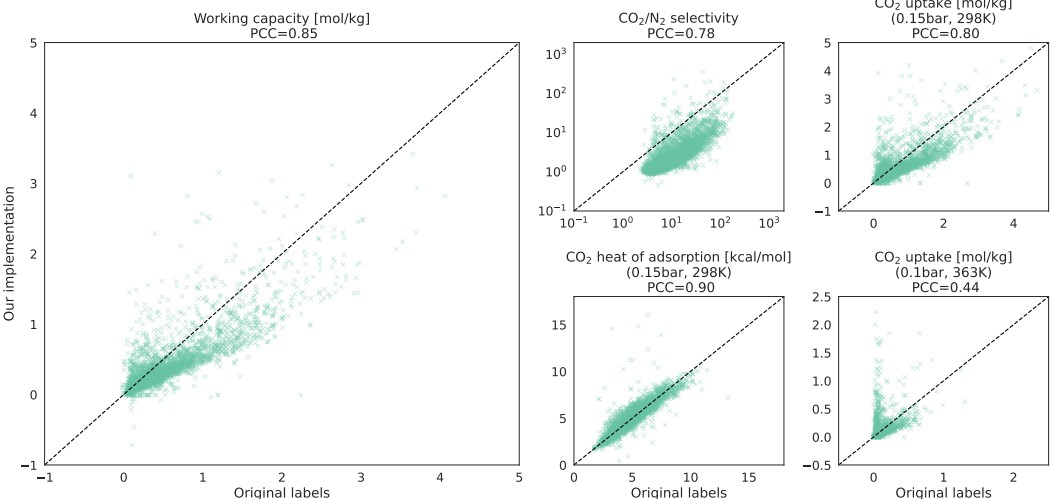

Figure 10: Benchmark GCMC results. PCC stands for "Pearson correlation coefficient".

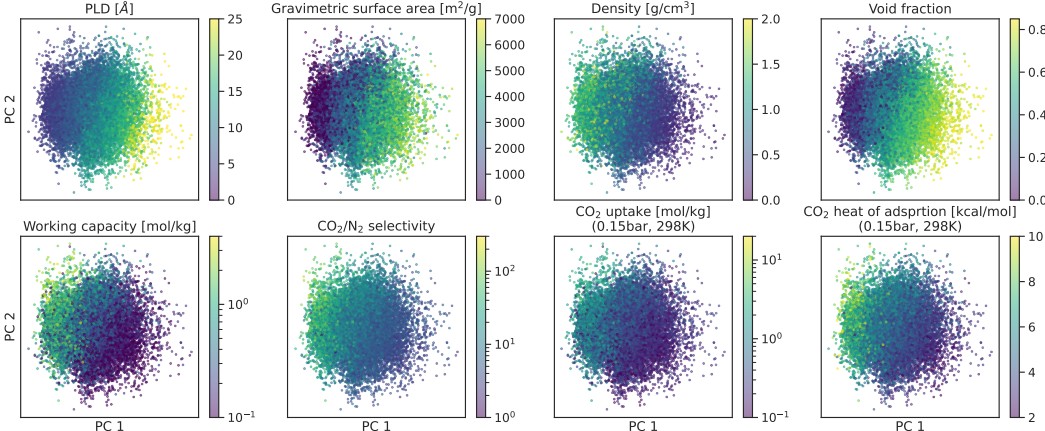

Figure 11: Principle component analysis of the MOFDiff latent space of the validation set, color-coded with various structural and gas adsorption properties.

Å [12]. We use 289k MOFs (95%) for training and the rest for validation. We do not keep a test split, as we evaluate our generative model on random sampling and inverse design capabilities.

**Molecular simulations for gas adsorption property calculations.** For faithful evaluation, we carry out grand canonical Monte Carlo (GCMC) simulations to calculate the gas adsorption properties of MOF structures. We implement the protocol for simulation of $CO_2$ separation from simulated flue gas with vacuum swing regeneration proposed in Boyd et al. [6] from scratch, using `egulp` to calculate per-atom charges on the MOF [35, 61] and `RASPA2` to carry out GCMC simulations [18] since the original simulation code is not publicly available. Parameters for $CO_2$ and $N_2$ were taken from Garcia-Sanchez et al. [21] and TraPPE [59], respectively. Under this protocol, the adsorption stage considers the flue exhaust a mixture of $CO_2$ and $N_2$ at a ratio of 0.15:0.85 at 298 K and a total pressure of 1 bar. The regeneration stage uses a temperature of 363 K and a vacuum pressure of 0.1 bar for desorption. Figure 10 shows the benchmark results of our implementation compared to the original labels of BW-DB, which demonstrate a strong positive correlation with our implementation underestimating the original labels by an average of around 30%. MOFDiff is trained over the original BW-DB labels and uses latent-space optimization to maximize the BW-DB property values. In the final evaluation, we use our re-implemented simulation code.

**Further details on molecular simulations.** Per-atom charges on the MOF were calculated with `egulp` using the `MEPO` parameter set and the default configuration. GCMC simulations were per-

Table 1: Carbon capture properties of top ten MOFDiff optimized samples and MOFs from previous literature, sorted by $CO_2$ working capacity.

| | $CO_2$ working capacity [mol/kg] | $CO_2/N_2$ selectivity | $CO_2$ uptake [mol/kg] (0.15 bar, 298 K) | $CO_2$ uptake [mol/kg] (0.1 bar, 363 K) | $CO_2$ heat of adsorption [kcal/mol] (0.15 bar, 298 K) | $CO_2$ heat of adsorption [kcal/mol] (0.1 bar, 363 K) |
|---|---|---|---|---|---|---|
| MOFDiff-1 | 4.89 | 197.66 | 7.05 | 2.16 | 10.13 | 10.05 |
| MOFDiff-2 | 4.86 | 65.17 | 6.57 | 1.71 | 9.39 | 9.00 |
| MOFDiff-3 | 4.03 | 39.55 | 5.08 | 1.05 | 7.85 | 8.44 |
| MOFDiff-4 | 4.03 | 26.21 | 4.85 | 0.82 | 9.05 | 8.41 |
| MOFDiff-5 | 3.87 | 1026.38 | 13.27 | 9.40 | 12.61 | 11.27 |
| Al-PMOF | 3.82 | 8.74 | 4.95 | 1.13 | 6.97 | 8.26 |
| MOFDiff-6 | 3.80 | 73.34 | 4.73 | 0.93 | 9.13 | 9.02 |
| MOFDiff-7 | 3.70 | 19.80 | 4.28 | 0.57 | 7.36 | 7.90 |
| MOFDiff-8 | 3.65 | 50.62 | 4.68 | 1.02 | 8.94 | 8.98 |
| MOFDiff-9 | 3.61 | 19.13 | 4.18 | 0.57 | 8.07 | 7.77 |
| MOFDiff-10 | 3.61 | 45.60 | 4.57 | 0.96 | 9.41 | 9.51 |
| InOF-1 | 3.11 | 9.26 | 3.43 | 0.32 | 7.61 | 6.69 |
| Ni-4PyC | 2.53 | 11.18 | 3.46 | 0.92 | 8.29 | 7.71 |
| MIL-53(Al) | 2.26 | 5.16 | 2.57 | 0.31 | 6.90 | 6.09 |
| MOOFOUR-1-Ni | 2.15 | 21.13 | 2.64 | 0.49 | 8.41 | 8.01 |
| UiO-66 | 2.11 | 19.15 | 2.70 | 0.59 | 7.82 | 8.72 |
| AlFu | 2.08 | 5.30 | 2.46 | 0.38 | 6.95 | 6.45 |
| SIFSIX-3-Cu | 1.22 | inf | 2.69 | 1.47 | 11.80 | 11.79 |
| NOTT-400 | 0.95 | 3.57 | 1.09 | 0.13 | 6.03 | 5.54 |
| MOF-14(Cu) | 0.88 | 3.11 | 1.02 | 0.14 | 5.93 | 5.66 |
| DICRO-3-Ni-i | 0.61 | 10.36 | 0.69 | 0.07 | 7.54 | 7.47 |
| MIL-100(Fe) | 0.53 | 3.61 | 0.63 | 0.10 | 5.82 | 6.88 |
| MIL-101 | 0.38 | 2.87 | 0.46 | 0.08 | 5.29 | 5.06 |
| CuBTC | 0.36 | 2.21 | 0.45 | 0.09 | 5.52 | 5.82 |
| DMOF-1 | 0.35 | 2.10 | 0.41 | 0.07 | 5.07 | 4.82 |
| ZIF-8 | 0.33 | 2.42 | 0.38 | 0.05 | 5.37 | 5.16 |
| MIL-125(Ti)-NH2 | 0.27 | 1.71 | 0.32 | 0.05 | 4.86 | 4.70 |
| MOF-5 | 0.09 | 1.02 | 0.12 | 0.03 | 3.34 | 3.11 |

formed with `RASPA2` using the default configuration unless otherwise noted. Charge-charge interactions were modeled with Ewald sums at a precision of $1 \times 10^{-6}$ J. Other interactions were modeled with the Lennard-Jones 12-6 potential using UFF parameters for the MOF atoms with the epsilon parameters scaled by 0.635, parameters from Garcia-Sanchez et al. [21] for $CO_2$, and TraPPE parameters for $N_2$. A 12.0 Åcutoff was applied to all interactions, with potentials shifted to zero at the cutoff radius. The minimum sized supercell was constructed for each MOF such that all lattice vectors were greater than 24.0 Åin length. The allowed Monte Carlo moves for gas atoms were identity change, swap, translation, rotation, and reinsertion at a likelihood ratio of 2:2:1:1:1, and the MOF atoms were held constant throughout the simulation.

Simulations were run for 2000 equilibrium cycles followed by 2000 production cycles, with the uptake of each gas calculated as the average loading over the 2000 production cycles as implemented in `RASPA2`. Similarly, each enthalpy of adsorption was calculated as the average internal energy of guest molecules within the MOF averaged over the 2000 production cycles as implemented in `RASPA2` and converted to heat of adsorption by changing the sign. Adsorption conditions were modeled using a mixture of $CO_2$ and $N_2$ at a partial pressure ratio of 0.15:0.85, an external temperature of 298 K, and an external pressure of 1 bar. Regeneration conditions were modeled using only $CO_2$, an external temperature of 363 K, and an external pressure of 0.1 bar. Working capacity was calculated as the difference in $CO_2$ uptake under adsorption and regeneration conditions. $CO_2/N_2$ selectivity was calculated as the ratio of each gas's respective uptake under adsorption conditions.

Figure 10 shows a benchmark that compares the gas adsorption labels obtained from BW-DB (original labels) and the labels obtained from our workflow (our implementation) for 5,000 randomly sampled MOFs from BW-DB. The Pearson correlation coefficient (PCC) is also reported for each property. We observe a strong positive correlation, while the working capacity is generally underestimated. Our model is trained with the original labels, and for property optimizing inverse design, we use a property predictor trained over the original labels. Our model still demonstrates significant property improvement (Figure 3), which demonstrates the robustness of our method under a shifted property evaluator.

**MOF latent space.** In Figure 11, we conduct a principle component analysis [34] to produce two-dimensional visualization of the MOFDiff latent space. The latent space exhibits smooth transitions for property values, indicating a smooth property landscape.

**Compare to literature MOFs.** In Table 1, we compare the top-ten MOFs generated by MOFDiff and 18 MOFs from previous literature [48, 13, 25, 6]. Notably, Al-PMOF was proposed in Boyd et al. 6, synthesized, and validated through real-world experiments.

**Software versions.** `MOFid-v1.1.0`, `MOFChecker-v0.9.5`, `egulp-v1.0.0`, `RASPA2-v2.0.47`, `LAMMPS-2021-9-29`, and `Zeo++-v0.3` are used in our experiments. Neural network modules are implemented with `PyTorch-v1.11.0` [20], `Pyg-v2.0.4` [58], and `Lightning-v1.3.8` [19] with CUDA 11.3.

**Code availability.** Source code will be released upon publication.

## F   Conclusion

We proposed MOFDiff, a coarse-grained diffusion model for metal–organic framework design. Our work presents a complete pipeline of representation, generative model, structural relaxation, and molecular simulation to address a specific carbon capture materials design problem. To design 3D MOF structures without using pre-defined templates, we derive a coarse-grained representation and the corresponding diffusion process. We then design an assembly algorithm to realize the all-atom MOF structures and characterize their properties with molecular simulations. MOFDiff can generate valid and novel MOF structures covering a wide range of structural properties as well as optimize MOFs for carbon capture applications that surpass state-of-the-art MOFs in molecular simulations.

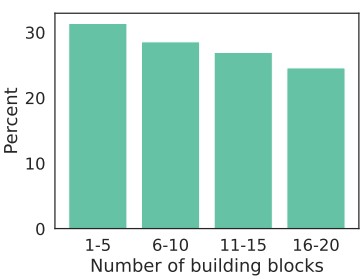

Figure 12: The percent of valid samples declines with more building blocks.

One limitation of MOFDiff is its generated samples have a lower validity rate when the size of the MOF becomes bigger. Figure 12 shows a declining validity percentage for samples of more building blocks. This result is unsurprising since a bigger MOF with more building blocks is inherently more complex. For a generated structure to be valid, the coordinates of every atom need to be correct, especially at every connection. The lattice parameters also need to be very accurate. Reformulating the diffusion process to enable the iterative refinement of the lattice parameters through the generation process and regularizing the diffusion process with known templates are two future directions to overcome this challenge.

Table 2: Hyperparameters for building block representation learning.

| Hyperparameter | Value |
|---:|:---|
| building block embedding dimension | 32 |
| GNN hidden layer dimension | 256 |
| projection dimension | 128 |
| # encoder GNN layers | 3 |
| radius cutoff | 20 |
| maximum number of neighbors | 50 |
| temperature ($\tau$) | 0.1 |
| $\beta_{\boldsymbol{b}}$ | 0.0001 |
| batch size | 512 |
| optimizer | Adam |
| initial learning rate | 0.0003 |
| learning rate scheduler | ReduceLROnPlateau |
| learning rate patience | 10 epochs |
| learning rate factor | 0.6 |

Table 3: Hyperparameters for MOFDiff.

| Hyperparameter | Value |
| --- | --- |
| latent dimension | 256 |
| GNN hidden layer dimension | 256 |
| # encoder GNN layers | 3 |
| # decoder GNN layers | 3 |
| radius cutoff | 4 |
| maximum number of neighbors | 24 |
| total number of diffusion steps ($T$) | 2000 |
| $\sigma_{\min}$ for coordinate diffusion | 0.001 |
| $\sigma_{\max}$ for coordinate diffusion | 10 |
| noise schedule for coordinate diffusion | $\sigma_t = \sigma_{\min}\left(\frac{\sigma_{\max}}{\sigma_{\min}}\right)^{\frac{t-1}{T-1}}$ |
| noise schedule for embedding diffusion | Hoogeboom et al. [29] |
| time step embedding | Fourier |
| time step embedding dimension | 64 |
| $\beta_{\mathrm{KL}}$ | 0.01 |
| batch size | 128 |
| optimizer | Adam |
| initial learning rate | 0.0003 |
| learning rate scheduler | ReduceLROnPlateau |
| learning rate patience | 50 epochs |
| learning rate factor | 0.6 |

