# OpenReview forum: "MOFDiff: Coarse-grained Diffusion for Metal-Organic Framework Design"
_NeurIPS.cc/2023/Workshop/AI4Science — NeurIPS2023-AI4Science Poster_

### Official Review · Reviewer_25CS · 2023-10-07

**Rating:** 9
**Confidence:** 5

**Review:**

The manuscript proposed a diffusion model based method for coarsed-grained MOF generation. From my perspective, the manuscript is clearly written and the method is sound. It bears high novelty as the manuscript is one of the first in deep learning MOF generation. The encoding and decoding process of building blocks using contrastive-learning pretrained GemNet can significantly reduce the computational burden of representing the MOF structure. It also allows the possibility of guided generation conditioned on desired MOF properties. The experiment results show that MOFs generated using MOFDiff can be very useful in gas adsorption tasks. The only question I have regarding this manuscript is about the overall motivation. There are works of representing MOFs using MOFid and Transformers, which can be modified to generate MOF through auto-regressively generating MOFid. Although such a method does not directly generate the 3D coordinate of atoms, the 3D MOF structure can still be obtained by computationally combine building blocks by specific topology. I am wondering what the authors' motivation behind choosing diffusion model instead of language models for MOF generations. Overall, I believe this manuscript is of very high quality to be on the AI4Science workshop.

---

### Official Review · Reviewer_q98Y · 2023-10-24
**great work**

**Rating:** 9
**Confidence:** 5

**Review:**

The authors develop a MOF design workflow based on generative diffusion model, which is the first attempt in the field afaik. The workflow involves multiple data-driven and physics-based methods and the designed candidates seem rather promising. all technical details are very clearly explained. I'd like to recommend acceptance of this manuscript.
A few comments from the perspective of a quantum chemist:
1. UFF is known to perform badly for organometallics and non-covalent interactions. There are better alternative FF out there for MOF. Or the authors can use semi-QM methods (e.g. xTB) which are way better while affordable.
2. In the assembling process, there seems to be no configurational sampling, which can be dangerous for linkers that contain flexible parts. I am also kinda skeptical of the coarse graining treatment for this reason.
3. The authors mention that the GCMC simulations are expensive -- why not train a model on GCMC simulations?
4. Adam is used to optimize CO2 working capacity. Could the authors also try using a global optimizer? I'd expect local optimizers to easily fall into local optima.
5. could the authors label some known popular MOF on the latent space PCA visualization for reference? It would be interesting to see where they are located.
6. how reversible and robust is the encoded representation? would like to see some testing

---

### Meta-Review · Area_Chair_rPfh · 2023-10-26

**Recommendation:** Accept (Poster)
**Confidence:** 4

**Metareview:**

Good paper.
Accept.